# Birth weight rather than birth length is associated with childhood behavioural problems in a Czech ELSPAC cohort

Lucie Ráčková[1], Daniela Kuruczová[1], Jiří Jarkovský[1], Julie Bienertová-Vašků[1,2]*

**1** Research Centre for Toxic Compounds in the Environment (RECETOX), Masaryk University, Brno, Czech Republic, **2** Department of Pathological Physiology, Masaryk University, Brno, Czech Republic

* julie.dobrovolna@recetox.muni.cz

**Data Availability Statement:** The analyzed dataset is not freely available for download as it contains potentially identifying and sensitive patient information. Datasets generated and/or analyzed in

## Abstract

Many physical and psychological characteristics are influenced by prenatal development. Some studies have located links between low birth parameters and behavioural problems, with the latter in turn associated with educational progress, career success, overall health, and subsequent life events. However, few studies have investigated whether this association also applies to children in the normal birth growth range. This study thus investigates the relationship between normal-range birth length, weight, and behavioural problems at the age of seven. We use data from the Czech part of the European Longitudinal Study of Pregnancy and Childhood (ELSPAC) cohort, which provides comprehensive insight into a post-communist country undergoing a period of economic transition. Childhood behavioural problems were measured in 1,796 children using the Strengths and Difficulties Questionnaire. Associations were modelled using weighted logistic regression. Birth weight was found to be linked to the total difficulties score, hyperactivity, and peer relationship problems sub-scales in a fully adjusted model while birth length was not significantly associated with any subscale in the fully adjusted model. We thus conclude that normal-range birth weight is associated with behavioural problems. It can therefore be assumed that the odds of behavioural problems and their consequences can be mitigated by preventive programs targeting pregnant women and children with lower but still normal weight.

## Introduction

The prenatal environment affects a child's biological and psychological development while also influencing their behaviour. Ever since the foetal origins of adult disease were observed by David Barker [1], various factors and their influence on child development have been studied extensively. The first studies were geographical and only circumstantial, focusing on the association between low birth weight or short birth length and the development of diseases in adult life (atherosclerosis, blood pressure, cognitive and affective disorders, depression, diabetes, dyslipidaemia, hypertension, stroke, coronary heart disease) [2–7]. As birth weight and birth length are not affected by postnatal development, they constitute suitable proxy measures

this study are available on reasonable request through the website of the Czech ELSPAC project: http://www.elspac.cz/index-en.php

**Funding:** This study has recieved support from RECETOX research infrastructure (Ministry of Education, Youth and Sports of the Czech Republic: LM2018121), Horizon 2020 Teaming 2 project (857560) and the Ministry of Education, Youth and Sports of the Czech Republic (CZ.02.1.01/0.0/0.0/17_043/ 0009632 and CZ.02.1.01/0.0/0.0/15_003/ 0000469).

**Competing interests:** The authors have declared that no competing interests exist.

of prenatal development. In fact, they are influenced by two evolutionary forces shaping the development of living organisms: nutrition and stress [8]. A number of stressors can alter foetal growth and result in birth weight lower than expected in accordance with the genetic potential of a given newborn [9]. Progress in neonatology enables low birth weight newborns–defined by the World Health Organization (WHO) as weighing less than 2,500 g [10]–to survive, thus allowing scientists to examine the impact of low birth weight on their health. Existing literature on this topic is extensive, ranging from the molecular mechanisms of prenatal programming mediated by glucocorticoids [11–14] and epigenetics [15] to evolutionary explanations striving to elucidate the ultimate causation [16–18]. Furthermore, it has been observed that some of the consequences of prenatal programming may be observed on children and can predict the abovementioned adult diseases. In addition, the prenatal environment can result in behavioural problems [19] which are predictors of mental health problems [20] but also extend to societal and economic problems [21].

Nevertheless, the relationship between birth weight, birth length and behavioural problems has remained a relatively unexplored area. Several studies have focused on low or very low birth weight children and are summarised in reviews or meta-analyses [22–24]. In general, these studies have found that low or very low birth weight is associated with behavioural problems as measured by validated questionnaires such as the Behavioural Assessment System for Children (BASC), the Child Behaviour Checklist (CBCL), the Infant-Toddler Symptom Checklist (ITSEA), and the Strengths and Difficulties Questionnaire (SDQ). However, as some of these studies combine premature and normal births, their results are heterogenous [24]; moreover, as they are based on specific populations of very low and low birth weight children, they have limited application with respect to the general population. Thus, in view of the missing information, some studies focused on exploring behavioural problems associated with birth weight on the full birth weight spectrum [25–30]. Because these studies differ in methodology, sample size and analytical approach, their comparability therefore hinges on these limitations. At present, a meaningful source of mutually comparable data can be reached by focusing on the common methodology employed by multinational cohort studies.

In 1985 the WHO launched the European Longitudinal Study of Pregnancy and Childhood (ELSPAC), the first geographically-based prospective population study of the prenatal and childhood environment conducted during the 1990s. The ultimate aim of ELSPAC was "to determine whether certain biological, environmental, social, psychological and psychosocial factors are associated with the survival, health and ability of the foetus, infant and child and to assess whether the same factors appear to be influential to a similar degree in each participating country" [31]. For this purpose, the WHO developed study design with specified enrolment process, methodology, and standardized procedures. Every cohort in each country had to follow this study design where also timepoints of administration were predefined [31]. Mothers were meant to be enrolled during pregnancy and then to receive multiple self-reported questionnaires in following years. The detailed information is provided in the research protocol of ELSPAC [31] and subsequent meeting reports [32–34]. Participating study centres included the Czech Republic, Slovakia (both former Czechoslovakia), Greece, the Isle of Man (UK), Russia, Ukraine and the United Kingdom. One aim of the ELSPAC cohort study was to provide data free of recall bias as well as an opportunity for repeated hypothesis testing between nations with different cultural profiles, which was enabled by the administration of the same questionnaires in similar timepoints. This strategy was designed to provide insight into causal mechanisms and the influence of cultural differences. However, due to a variety of reasons only cohorts from the Czech Republic, the Isle of Man, Slovakia, Ukraine and the United Kingdom succeed in measuring children up to the age of seven and beyond [34].

From the above-mentioned cohorts, data from the Czech part of the ELSPAC study provide an exceptional opportunity for investigating child development from a longitudinal perspective in a post-communist Central European country during a period of economic transition. As Sobotka et al. [35] elaborated, the transition from totalitarian regime to democratic institutions and the market economy resulted in a complex demographic transformation. These include decline of period fertility rates, increase of children born outside marriage, postponement of, marriages and childbearing. This was described as a form of crisis behaviour in reaction to unfavourable economic factors, such as high unemployment, inflation, economic crisis, and uncertainty. The transition left a clear imprint on the behaviour of people in the 1990s [35], and possibly on the mothers in our Czech ELSPAC cohort. The longitudinal design ensures a low probability of recall bias. In addition, findings established by the study may be compared with methodologically and temporally concordant studies conducted elsewhere within the ELSPAC framework.

This study aims to investigate the relationship between birth weight, birth length and behavioural problems using the Strength and Difficulties Questionnaire (SDQ) at age seven using data from the Czech ELSPAC cohort. Our hypotheses are, that lower birth weight and birth length will be related to the higher scores of the childhood behavioural problems and that the results will be similar as those from the concurrent ELSPAC cohort from United Kingdom, referred to as ALSPAC.

## Materials and methods

### Subjects

Datasets originate from ELSPAC, a prospective study initiated by the WHO Regional Office for Europe in 1985. The study included 40,000 children from six European countries and focused specifically on biological, psychological, social, economic and environmental factors influencing the health of children and adolescents [31]. Table 1 shows comparison of the key timepoints and number of children (newborns and at 7 years). This data is available from the individual Cohort profiles or by WHO reports. From among the participating countries (Table 1), only information from the Czech Republic was used here.

**Table 1. Basic information on countries participating in ELSPAC.**

|  | Date of delivery | No. of newborns | No. of children at 7 years | 7 years follow-up | Reference |
|---|---|---|---|---|---|
| **Czech Republic** | March 1991–June 1992 (Brno), April 1991–June 1992 (Znojmo) | 7,589 | 3,311 | March 1999 (Brno) | [36] |
| **Greece** | – | 6,441 | – | – | [33] |
| **Isle of Man (UK)** | January 1991–June 1992 | 1,314 | 1,150* | June 1998 | [37] |
| **Russia** | – | 5,336 | – | – | [33] |
| **Slovakia** | – | 3,005 | – | September 2000 | [33] |
| **Ukraine** | December 1992–June 1994 (L bank Kyiv, Dniprodzerzhynsk), August 1993–August 1996 (Mariupol) | 7,271** |  | September 2000 | [33] |
| **United Kingdom** | April 1991–December 1992 | 13,761 | 8,290 | August 1998 | [33, 38] |

UK, United Kingdom

* number of subjects from original cohort, does not include follow-up cohort

** sum of all participating cities

The Czech ELSPAC study includes data on participants born between March 1991 and June 1992 in Brno (approximately 0.4 million people) and Znojmo (~ 0.1 million people) in the South Moravian region of former Czechoslovakia. Mothers were recruited during pregnancy. The entire Czech ELSPAC population consists of 7,589 children. The demographic parameters indicate that the cohort is a representative sample of the Czech population in that time. Detailed information on population characteristics, study design and ethic approval is provided elsewhere [36].

## Data

The analysed sample consisted of all singleton infants born at term ($\geq$37 completed weeks gestation) with available information on birth length, birth weight. From 7,589 children of the whole Czech ELSPAC cohort, 5,630 had all the above-mentioned data available and consist our study sample.

## Birth length and weight

Data on birth length, birth weight and gestational age were obtained from obstetrics records. Information on birth length and birth weight was collected from medical records by trained medical personnel in accordance with standard operating procedures. Gestational age was calculated using the reported date of the mothers' last menstrual period or based on ultrasound scan results.

## Behavioural problems

Behavioural problems were evaluated on the basis of data from the parental version of the Strengths and Difficulties Questionnaire (SDQ). The SDQ is widely used in epidemiological studies [37–39] and its validity and reliability were previously assessed in the case of the English language version [40, 41]. The psychometric proprieties of the SDQ questionnaire in the ELSPAC cohort are reported elsewhere [42]. The SDQ questionnaire consists of 25 items focusing on psychological attributes, scored 0 to 2 ("not true" to "certainly true"). The answers on the items should reflect child's behaviour over the last six months. These 25 items are divided by 5 into 5 subscales: emotional symptoms, conduct problems, hyperactivity/inattention, peer relationship problems and prosocial behaviour, each with scores ranging from 0 to 10. Total difficulties scores are calculated by summing the first four listed subcategories, with resulting scores ranging from 0 to 40. High scores indicate more problems, except for the prosocial subscale which is reversed. Questionnaires were reported by mothers for their children at age 7.

## Confounders

To assess for possible confounders the information on the following factors was included: sex of the child, maternal smoking during pregnancy, maternal age at birth, socioeconomic factors, pre-pregnancy body mass index (BMI), maternal depression and anxiety, and single-parent household. Data were collected using self-administered P2 questionnaires (available here: [43]) filled out by mothers during pregnancy. Maternal smoking was categorized as follows: never smoked, not currently smoking but smoked before pregnancy, currently smoking. Maternal age at birth was categorized as <20, 20–24, 25–29, 30–34, 35–40, 40<. Socioeconomic markers were measured by assessing maternal education which was subdivided into three categories according to the International Standard Classification of Education [44]: low–elementary school and/or vocational qualification, medium–completed secondary school at

the age of 18 or equivalent, high–academic qualification reached at 21 years of age or later. Pre-pregnancy BMI was calculated using Quelet's index: weight (kg)/height (m2) and categorized as <18, 18–24.9, 25–29.9, 30<. The single-parent household value was binomially categorised as true or false. Unfortunately, because of the lack of complete data, this study was not able to include information on confounders of parity, household overcrowding, housing tenure, car ownership and children's IQ.

## Statistical analysis

Data analysis was performed using statistical software R, version 3.6.1. [45]. Subjects with full information on birth length, birth weight, behavioural problems and all covariates were further used in the model calculation as the Included dataset (n = 1,796) while the Excluded dataset (n = 3,834) was comprised of all other subjects, i.e. singletons who completed 37 weeks' gestation and had at least one missing data point. Exploratory data analysis was performed on both datasets.

In order to examine the relationship between birth weight, birth length and childhood behaviour, a weighted logistic regression model was fitted. The binary response was defined as the highest tercile of the SDQ score at age 7 for all four problem subscales and as the lowest tercile for the prosocial behaviour. Birth length and birth weight were expressed as z-scores calculated according to WHO Growth Chart Standards and WHO Reference 2007 [46]. Therefore, the odds ratio can be interpreted as an increase or decrease in the odds of behavioural problems corresponding to one standard deviation (SD).

The weighted logistic regression method was selected to account for the somewhat different proportion of socio-economic characteristics in the Included and Excluded dataset [47]. The first step in the construction of weighted logistic regression was the calculation of weights. Using all subjects included in the analysis, a logistic regression model was constructed to predict that a given subject belongs to the Included dataset based on the following covariates: single parenthood, maternal education, maternal age and sex of the child. Reversed values of predicted probabilities from the model were subsequently used as weights. Next, four weighted logistic regression models were fitted. The first (unadjusted) model explored the relationship between behavioural problems and birth length/weight. The second model expanded the first by including the sex of the child and gestational age. Birth length/weight was added in the third model while the fourth (fully adjusted) model included all confounders, i.e. maternal education, maternal age, pre-pregnancy BMI, single parenthood and smoking.

To confirm that the dichotomization of response variable and use of logistic regression did not bias the results, we fitted all models also as linear regression models with numeric SDQ score as response variable.

## Results

### Birth length and weight in ELSPAC children

The source dataset for the logistic regression model consisted of 1,796 singleton children (906 boys and 890 girls). Their mean birth weight was 3.35 kg (standard deviation (SD) 0.44 kg) and mean birth length was 50.43 cm (SD 1.99 cm). These values did not differ much from dataset excluded from the model, which included 3,834 children with a mean birth weight of 3.33 kg (SD 0.43 kg) and mean birth length of 50.37 cm (SD 1.97 cm). Detailed results with data on girls, boys and the Excluded dataset can be compared in Table 2.

Mothers in Included dataset were on average 25.98 years old (SD 4.84 years) when giving birth and their pre-pregnancy BMI was 22.05 (SD 3.30). Of those, 62.31% never smoked, 31.07% used to smoke and 6.63% smoked during pregnancy. Regarding educational status,

**Table 2. Mean and standard deviation for birth length and birth weight for children in the Czech ELSPAC cohort.**

| | Included | | | | | | Excluded | |
| --- | --- | --- | --- | --- | --- | --- | --- | --- |
| | Analysis set | | Boys | | Girls | | Total | |
| | n = 1,796 | | n = 906 | | n = 890 | | n = 3,834 | |
| | Mean | SD | Mean | SD | Mean | SD | Mean | SD |
| Birth weight (kg) | 3.35 | 0.44 | 3.43 | 0.44 | 3.27 | 0.41 | 3.33 | 0.44 |
| Birth length (cm) | 50.43 | 1.99 | 50.99 | 1.95 | 49.95 | 1.91 | 50.37 | 1.97 |

33.57% completed primary/vocational education, 45.99% completed secondary and 20. 43% completed tertiary education.

### SDQ score at age 7

In the included dataset, the mean of the total SDQ scores measured at age 7 was 9.06 (SD 4.77). The results of the Excluded dataset are similar to those of the Included dataset, as shown in Table 3.

### Birth weight and SDQ at 7y

The results of the weighted logistic regression examining the relationship between birth weight and behavioural problems are presented in Table 4. They indicate that the results on total difficulties and hyperactivity subscales were statistically significant across all models, i.e. from the unadjusted to the fully adjusted. In the latter, one SD increase in birth weight was associated with a 20% (95% confidence interval: 0.71, 0.92) reduction of odds for having the total difficulties score in the highest tertile. Similarly, the odds for the hyperactivity subscale were reduced by 17% (95% confidence interval: 0.71, 0.97). Another significant result from the fully adjusted model was between birth weight and peer relationship problems, where one SD increase in birth weight reduced the odds of scoring in the highest tertile of peer relationship problems by 15% (95% confidence interval: 0.75, 0.97). Full report on statistical models for birth weight along with model quality measures is reported in (S1 Table).

### Birth length and SDQ at 7y

For birth length, one SD increase was associated with a decrease in the odds of scoring in the highest tertile for conduct problems by 7% (95% confidence interval: 0.87, 0.99) in the unadjusted model and by 12% (95% confidence interval: 0.81, 0.94) in the model adjusted for gender and gestational age. However, after adjustment for birth weight or after full adjustment,

**Table 3. Mean and standard deviation for scores on the strengths and difficulties questionnaire at age 7 in Czech ELSPAC cohort.**

| | Included | | | | | | Excluded | | |
| --- | --- | --- | --- | --- | --- | --- | --- | --- | --- |
| | Analysis set | | Boys | | Girls | | | | |
| | n = 1,796 | | n = 906 | | n = 890 | | | | |
| | Mean | SD | Mean | SD | Mean | SD | Mean | SD | n |
| **Total difficulties** | **9.06** | **4.77** | **9.61** | **4.77** | **8.51** | **4.70** | **9.27** | **5.08** | **566** |
| Hyperactivity | 3.58 | 2.17 | 3.87 | 2.26 | 3.27 | 2.02 | 3.66 | 2.22 | 607 |
| Conduct problems | 1.87 | 1.54 | 2.09 | 1.59 | 1.64 | 1.45 | 1.95 | 1.53 | 617 |
| Emotional problems | 1.88 | 1.72 | 1.83 | 1.70 | 1.93 | 1.74 | 2.01 | 1.92 | 607 |
| Peer problems | 2.04 | 1.37 | 1.82 | 1.49 | 1.66 | 1.45 | 1.81 | 1.60 | 586 |
| Prosocial behaviour | 6.74 | 1.52 | 7.44 | 1.73 | 7.99 | 1.59 | 7.77 | 1.75 | 597 |

**Table 4. Weighted models for the association of birth weigth and childhood behavioural problems measured using strength and dificulties questionnaires at age 7 in the Czech ELSPAC cohort.**

| Outcome (highest/lowest) tertile of behavioural problems | Unadjusted model | | | Adjusted for gender and gestational age | | | Adjusted for birth length | | | Fully adjusted model | | |
|---|---|---|---|---|---|---|---|---|---|---|---|---|
| | OR | 95% CI | | OR | 95% CI | | OR | 95% CI | | OR | 95% CI | |
| **Total difficulties** | **0.90** | **0.84** | **0.98** | **0.85** | **0.78** | **0.93** | **0.80** | **0.70** | **0.91** | **0.80** | **0.71** | **0.92** |
| **Hyperactivity** | **0.89** | **0.81** | **0.98** | **0.81** | **0.73** | **0.91** | **0.82** | **0.71** | **0.96** | **0.83** | **0.71** | **0.97** |
| **Conduct problems** | **0.92** | **0.85** | **1.00** | **0.85** | **0.78** | **0.92** | 0.90 | 0.79 | 1.02 | 0.90 | 0.79 | 1.02 |
| **Emotional problems** | 1.02 | 0.94 | 1.10 | 0.99 | 0.91 | 1.08 | 1.02 | 0.90 | 1.15 | 1.04 | 0.91 | 1.18 |
| **Peer problems** | 0.96 | 0.89 | 1.04 | 0.92 | 0.84 | 1.01 | **0.86** | **0.76** | **0.98** | **0.85** | **0.75** | **0.97** |
| **Prosocial behaviour** | 0.95 | 0.87 | 1.03 | 0.91 | 0.83 | 1.00 | 0.94 | 0.82 | 1.08 | 0.94 | 0.82 | 1.08 |

the odds were non-significant. An additional association was found for hyperactivity in the model adjusted for gender and gestational age. Nevertheless, after adjustment for birth weight and after full adjustment, the association became insignificant. Results for other subscales and for the total difficulties score were not significant. Specific numbers can be found in Table 5. Full report on statistical models for birth length along with model quality measures is reported in (S1 Table).

## Discussion

The current study used unique data from a Central European population undergoing a period of socioeconomic transition to explore the relationship between birth weight, birth length and behavioural problems in children at age seven. We found that birth weight was negatively associated with the total difficulties score and subscales of hyperactivity. In contrast, birth length was only associated with conduct problems in one of our models; following adjustment for birth weight, the association weakened.

Our results contrast with those of the temporally and methodologically concordant UK-based ALSPAC cohort [48] which differs in the number of participants and the gene pool as well as in socioeconomic and other factors emerging from differences in political systems [34, 35, 37]. On the other hand, similarities between the two cohorts include the associated subscales and the direction of association: the total difficulty score, hyperactivity and also conduct problems which reached only weak association in our study. However, while our study found these associations with birth weight, ALSPAC established an association with birth length. For birth length model, we found an association only with subscales of hyperactivity and conduct problems; however, after adjustment for birth weight or after full adjustment the association

**Table 5. Weighted models for association of birth length and childhood behavioural problems measured using strength and dificulties questionnaires at age 7 in the Czech ELSPAC cohort.**

| Outcome (highest/lowest) tertile of behavioural problems | Unadjusted model | | | Adjusted for gender and gestational age | | | Adjusted for birth weight | | | Fully adjusted model | | |
|---|---|---|---|---|---|---|---|---|---|---|---|---|
| | OR | 95% CI | | OR | 95% CI | | OR | 95% CI | | OR | 95% CI | |
| **Total difficulties** | 0.97 | 0.90 | 1.04 | 0.94 | 0.87 | 1.01 | 1.07 | 0.96 | 1.19 | 1.09 | 0.98 | 1.22 |
| **Hyperactivity** | 0.93 | 0.86 | 1.01 | **0.88** | **0.80** | **0.96** | 0.98 | 0.86 | 1.11 | 0.99 | 0.87 | 1.12 |
| **Conduct problems** | **0.93** | **0.87** | **0.99** | **0.88** | **0.81** | **0.94** | 0.93 | 0.84 | 1.03 | 0.95 | 0.86 | 1.06 |
| **Emotional problems** | 1.00 | 0.93 | 1.07 | 0.98 | 0.91 | 1.05 | 0.96 | 0.87 | 1.07 | 0.96 | 0.87 | 1.07 |
| **Peer problems** | 1.01 | 0.94 | 1.09 | 0.99 | 0.92 | 1.07 | 1.08 | 0.97 | 1.20 | 1.09 | 0.98 | 1.22 |
| **Prosocial behaviour** | 0.95 | 0.89 | 1.03 | 0.93 | 0.86 | 1.01 | 0.96 | 0.86 | 1.08 | 0.97 | 0.87 | 1.09 |

weakened (for conduct problems) or became insignificant (for hyperactivity). This may have been caused by our smaller sample size compared to ALSPAC.

The reason for this inconsistency in the associated parameters may be due to the inter-population variability of prenatal growth, since the interpopulation variability in postnatal height was reported previously [49]. While ethnic differences in prenatal growth have been examined previously [50–52], Villar et. al reported that when maternal nutritional and health needs are fulfilled these differences only account for several percent of variability [53]. Another reason could be the different timing of prenatal stress in these cohorts. The results from ALSPAC cohort [48] indicate that the established associations might be due to disturbances in the first two trimesters, a period of cellular hyperplasia where the foetus lengthens more rapidly, which may result in the reduction of both birth length and birth weight. Our results show an association only with low birth weight which may have been caused by restrictions in the final trimester which is a period of the greatest weight gain due to cellular hypertrophy [9]. Nevertheless, our outcome is in good agreement with Yang et al. [54], who also found an association between birth length and hyperactivity, which diminished after adjustment for birth weight.

A comparison of our findings with other studies focusing on the full birth weight spectrum confirms that birth weight is negatively associated with the total difficulties score and hyperactivity subscale score measured using SDQ questionnaires [26, 30, 54]. These results expand previous findings of the relationship between very low birth weight and hyperactivity [55]. It was also suggested that the higher total difficulties score results from additive minor changes in all subdomains but mainly from emotional problems in girls and hyperactivity in boys [30]. The sex differences in the influence of the birth weight on the specific subscales was observed in some studies. They have found higher scores of hyperactivity only in boys and higher scores of emotional problems in girls [27, 30]. Our study neither tested for sex differences nor did we find a significant influence on the subscale of emotional problems. The significance of the established association between birth weight and peer relationship problems was consistent with two other studies which also established this problem in girls [26, 27] and in children with birthweight lower for gestational age [54]; however, this association was not found in other study [30]. A weak negative relationship between conduct problems in our study is comparable with the results of other studies which similarly found such association [26, 54]; however, it differs from one study which did not establish any such relationship [30]. The absence of an association between birth weight and emotional problems is in good agreement with two studies [26, 54] but at odds with an additional one [30]. These inconsistencies may be due to cultural differences in question comprehension, willingness to answer truthfully or in the tendency to misrepresent results and in the perception of what constitutes good or bad behaviour [56].

There is a possible connection between prenatal stress, birth weight or length and behavioural problems in childhood. It is known that birth length and weight could be influenced by the influence of prenatal stress on the prenatal environment [8]. Several studies also found that prenatal stress can also impact childhood behaviour. For example, Štěpaníková et al. [57] reported a positive association between maternal prenatal stress and childhood behavioural problems in the same Czech ELSPAC cohort. A study by MacKinnon et al. of the ALSPAC cohort found an association between prenatal stress, hyperactivity and conduct problems in children [58]. It seems possible that a stressful prenatal environment is one of the factors resulting in lower birth weight and externalizing problems, such as hyperactivity and conduct problems. There are several possible mediating factors which may underlie the relationship between prenatal environment and childhood behaviour. During foetal development, the foetus reacts to external stimuli by changing its development pattern to prepare the child for the postnatal environment [59]. Altered developmental pattern of the foetuses' brain, more

specifically the regions of brain stem, limbic and cortical areas, may consequently alter the child's future behaviour [60–62]. Another possible mediating factor might be effortful control [63], or verbal skills, which were found to negatively mediate aggressive and destructive behaviour [64]. Overall, however, mediating factors do not appear to be fully explored. From an evolutionary perspective, the children with symptoms of hyperactivity and conduct problems are more sensitive to dangerous signals as well as more impulsive to explore their environment, which could improve their odds of countering threats as well as their overall chances of survival [59, 65].

Our findings provide support for the conceptual premise that the unfulfilled growth potential of the foetus leads to an increased tendency to express externalizing problems, which may in turn be reflected in the quality of adult life depending on the context in which the individual subsequently lives. They raise important implications for developing preventive programmes which should mitigate this effect. The proper prenatal development of children may be supported by pregnancy programmes. Behavioural problems in children can be mitigated by improved effortful control [66], by specific parenting styles and positive family relationships [67–72] or by the attitude of teachers [73].

The main strength of our study stems from its timing, i.e. from employing data collected during the socioeconomic transition of the Czech Republic from command to market economy, and from the quality of data on birth weight and birth length, obtained by trained medical personnel. On the other hand, study limitations include sampling size or selection bias due to drop out during follow-up [36]. Another source of bias may lie in errors introduced during the use of measurement tools and techniques for birth length and weight acquisition. Furthermore, when comparing our results to the methodically and temporally concordant ALSPAC study, bias may be introduced due to nature of parent reporting assessment of behavioural problems and deviations in the cross-cultural validity of the SDQ questionnaire. One study by Essau et al. [74] identified considerable differences in internal consistency between European versions of the SDQ questionnaire. To the best of our knowledge, there is no study examining the validity between the Czech and UK version. Furthermore, we did not adjust for all possible confounders (such as hypertension or parenting style) or genetic factors. Lastly, because most of the ELSPAC cohorts have not published or provided the data the comparison with other cohorts was not possible [75].

The key finding emerging from this study is that our analysis of the ELSPAC cohort, conducted at seven years of age, revealed an association between birth weight and behavioural problems measured by SDQ. More specifically, low birth weight was found to increase the odds of high scores in total difficulty problems, hyperactivity, conduct problems and peer relationship problems. This outcome is in agreement with most literature but somewhat contrary to ALSPAC cohort findings, which indicate a more relevant association between behavioural problems and birth length. Therefore, we conclude that behavioural problems in children are associated with birth weight in general and that they may be caused by growth restrictions during the entire pregnancy period.

## Supporting information

**S1 Table. Weighted models: Complete results for models of SDQ (total score and individual subscales) at 7 years.**
(XLSX)

**S1 File.**
(XLSX)

## Acknowledgments

The authors of this study would like to thank the participating famillies and the gynaecologists, paediatricians, school heads and class teachers who took part. Our thanks go to Dr. Lubomír Kukla, PhD who was the ELSPAC national coordinator 1990–2012, Lenka Andrýsková, PhD and Jiří Jarkovský, PhD from ELSPAC infrastructure and also to the entire ELSPAC team. We thank Mgr. David Konečný for writing assistance, language editing and proofreading.

## Author Contributions

**Conceptualization:** Lucie Ráčková, Julie Bienertová-Vašků.

**Data curation:** Daniela Kuruczová, Jiří Jarkovský.

**Formal analysis:** Daniela Kuruczová.

**Investigation:** Daniela Kuruczová, Julie Bienertová-Vašků.

**Methodology:** Jiří Jarkovský.

**Project administration:** Lucie Ráčková.

**Writing – original draft:** Lucie Ráčková.

**Writing – review & editing:** Daniela Kuruczová, Julie Bienertová-Vašků.

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
