## [Decision Letter · Decision Letter 0]

25 Mar 2021

PONE-D-21-02707

Birth weight rather than birth length is associated with childhood behavioural problems in a Czech ELSPAC cohort

PLOS ONE

Dear Dr. Bienertová-Vašků,

Thank you for submitting your manuscript to PLOS ONE. After careful consideration, we feel that it has merit but does not fully meet PLOS ONE’s publication criteria as it currently stands. Therefore, we invite you to submit a revised version of the manuscript that addresses the points raised during the review process.

This manuscript is quite well written and the results are of interest (novelty not being a publication criteria for this journal). However both reviewers have major concerns about the manuscript, especially in relation to the weighting used in the statistical analysis. If the authors decide that they will revise the manuscript modifying this (or at least rebutting the concerns robustly) is key. Reviewer 1 also has other concerns about the statistical analysis which also require robust responses, and reviewer 2 needs clarification of a number of different points, which I agree with.

Currently I feel that the data availability is not compatible with publication in this journal. At a very minimum more information is needed as to how other investigators may gain access to the data.

We look forward to receiving your revised manuscript.

Kind regards,

Clive J Petry, PhD

Academic Editor

PLOS ONE

Journal Requirements:

3. Thank you for including your ethics statement that the research was approved by ELSPAC Law and Ethics Committee and local research ethics committees.

a. Please amend your current ethics statement to include the full name of all of the ethics committee/institutional review board(s) that approved the ELSPAC study.

4. Thank you for stating the following after the Acknowledgments Section of your manuscript:

'This study has recieved support from RECETOX research infrastructure (Ministry of Education, Youth and Sports of the Czech Republic: LM2018121), Horizon 2020 Teaming 2 project (857560) and the Ministry of Education, Youth and Sports of the Czech Republic (CZ.02.1.01/0.0/0.0/17_043/ 0009632 and CZ.02.1.01/0.0/0.0/15_003/ 0000469).'

'NO - The funders had no role in study design, data collection and analysis, decision to publish, or preparation of the manuscript.'

Reviewers' comments:

Reviewer's Responses to Questions

**Comments to the Author**

1. Is the manuscript technically sound, and do the data support the conclusions?

Reviewer #1: Partly

Reviewer #2: Partly

2. Has the statistical analysis been performed appropriately and rigorously? 

Reviewer #1: No

Reviewer #2: No

3. Have the authors made all data underlying the findings in their manuscript fully available?

Reviewer #1: No

Reviewer #2: No

4. Is the manuscript presented in an intelligible fashion and written in standard English?

Reviewer #1: Yes

Reviewer #2: Yes

5. Review Comments to the Author

Reviewer #1: This study examines the relationship between birth weight, birth length and childhood behaviour. I think the paper has some merits but I also have some key concerns, which I detail below. I wish the authors all the best taking his work forward.

A first major issue that emerges from my own reading of the paper concerns the methods used. Why is a binary response constructed for each behavior problem? Why is the original variable not used? I do not understand why it is necessary to dichotomize the variable with the corresponding loss of information that it implies. I think that you can estimate an OLS model. Perhaps a fixed effects model will be better in order to control the effect of unmeasured characteristics of the family (biased estimates). I would like to see the results of a such model.

Moreover, the decision to weight the data is not well justified. It is said that the method was selected to account for the somewhat different proportion of socio-economic characteristics in the Included and Excluded dataset. I understand that there is some pattern that leads to non-response, such as that families with lower socio-economic status tend to respond less or, perhaps it is a problem of the sample….. It is necessary to better specify the use of weighting, since it is true that it reduces bias but weighted often increases variances of the estimates and its use must be well justified.

Four weighted logistic regression models were fitted, one for each behavior problem and separated by birth weight and birth length. I understand that Tables 4 and 5 show the OR for the relationship of birth weight and birth length respectively, but it does not show the significance of the rest of the variables included in each model or any measure of the statistical quality of the model to compare, for example the AIC, BIC, ...

In the other hand, subjects with full information were in the Included dataset, and all other subjects, who had at least one missing data point, were in the Excluded dataset. I understand that these two databases do not have subjects in common but the Included dataset have 1796 subjects and the Excluded 3834, and that differs from the 3311 total subjects (Table 1). Also, line 198 talks about mothers in both datasets, so, I need a better explanation.

Regarding the Strengths and Difficulties Questionnaire, the explanation provided is not very clear. First it is said that it contains 25 items scored 0 to 2, what does each value correspond to? Them, these are separated into five subscales measured from 0 to 10. How is it related to the previous score? I understand that each subscale contains different items to value, isn’t it? It is necessary to improve.

Finally, last major issue concerns the contribution of your study with the extant knowledge. While I agree that the topic is interesting, I think that these findings do not provide anything new. Maybe it would be interesting to analyze differences between sex or between some socioeconomic characteristics of the family.

Lastly, some lower priority comments:

- Explain why behaviour problems are analysed at age 7 and not at age 8 or 6 for example.

- The first time the acronym WHO is used is on line 46, but its definition is on line 72.

- When we talk about association between variables, we can say significant or non-significant association with a certain risk. The term “weak association” is not adequate.

- There is an error on line 238: 0,97 would have to be 0,79

Reviewer #2: In the current study, the authors investigate whether anthropometric measures at birth are associated with later behavioural outcomes in childhood. While the study has some merit, I believe that some changes need to be made to the current manuscript before it is suitable for publication. These changes broadly include giving more context and background around the time period at which the data were collected and clearly articulating the stress that this may have caused to the pregnant mother and developing foetus.

I also have some concerns around the use of inverse-probability weights to account for missing data rather than measurement invariance and the interpretation of some negligible effects as significant. Detailed feedback is given below.

Abstract

1. Give the sample size here

2. Page 2, Line 25 should say “the Strengths and Difficulties Questionnaire” rather than “a Strengths and Difficulties Questionnaire”.

Introduction

1. Page 2, line 43 – what is meant by “evolutionary” forces regarding nutrition and stress? An explanation of this is needed here.

2. On page 3, lines 66 – 68 – the authors state that some studies focus on exploring behavioural problems in children belonging to the full birth weight spectrum, but then say that the studies differ in methodology, sample size and analytical approach. What is the extent of the differences and why might it pose a problem? It’s quite common for studies to vary in methodology, sample size and even analyses, but a robust effect should be apparent across sound methods. For example, one study may have 200 participants, and another may have 20,000. I would still consider the sample size of 200 acceptable if other methods were sound, and not necessarily a limitation. Essentially, this section is quite vague and needs more information.

3. On page 4, the authors note that the ELSPAC study centres all employ a common methodology. What exactly is meant by this – was the recruitment procedure the same? The questions asked the same?

4. On page 4, the authors also mention that the aim of the study was to provide data free of recall bias. How exactly was this bias reduced? For example, the SDQ still requires some level of recollection. Was the study only asking about events in the 2 weeks prior, month prior, etc.? Were women recruited during pregnancy rather than postnatally?

5. The authors mention that the Czech Republic, which the cohort was from, was a post-communist country undergoing a period of economic transition at the time. I think some more context around the importance of this period is needed and why it might be relevant to child development. Was it a period of stress for the mothers? This needs to be clearly articulated, and the context will be informative for non-European readers.

6. The last sentence on page 4 (“we then compare the results with outcomes obtained using the concurrent ELSPAC cohort from United Kingdom, referred to as ALSPAC (33) as well as other studies employing Strength and Difficulties Questionnaires (SDQs) to measure behavioural problems”) sounds like data from ALSPAC and other studies are being analysed and compared to ELSPAC data within the results. I don’t think this sentence is necessary, as it is to be expected that these comparisons would be made within the discussion. Rather, I would suggest stating hypotheses instead (perhaps based on what some of these studies have found).

Methods

1. The authors note at the start of page 6 that detailed information about the study is available elsewhere. While detailed information certainly isn’t expected, I still think that a brief description of the study is needed. E.g. was the cohort recruited before birth? Is it representative of other births in the Czech Republic? The authors also mention “ethical issues” – is this referring to ethics approval for the study, or were there specific ethical concerns around the study’s methodology/data collection/handling of participants?

2. Page 7, line 150 – give a reference for the International Standard Classification of Education.

3. In the statistical analysis section, I would recommend that the authors give the Ns for the included and excludes subjects.

4. What was the rationale for the use of inverse probability weighting (IPW) rather than multiple imputation (MI) to account for the impact of missing data? While IPW does adjust for differences between those included and those excluded in the sample, MI has the benefit of accounting for missing data while not reducing sample size. A combined approach could also be used here.

Results

1. Page 11 – were the differences between boys and girls tested statistically? If so, indicate that these were statistically significant p-values. If not, then I’m not sure we can really say that males and females scored differently without statistical testing.

2. I would recommend indicating significant effects in the tables, particularly Tables 4 and 5. Perhaps bolding the OR and C.I. numbers for significant effects.

3. For the birthweight models, I don’t see the value in interpreting effects where the confidence interval contains 1, even if considered weak. My concern is that it can be misleading – indicating a more meaningful effect than what is actually present.

Discussion

1. As stated with regard to the results, the effects were very weak for models with conduct problems and prosociality as the outcome, with the C.I.s containing 1. Interpreting these models as significant is a bit misleading, suggesting that there may be a more meaningful effect.

2. Page 15, line 269 – what is meant by “genetic environment”?

3. Page 15, lines 274-275 – the statement “though this was no longer apparent in our birth weight model adjusted for birth length” isn’t needed, as it is redundant given the sentence that follows.

4. For the first two sentences on page 16, it’s not clear how exactly this relates to why there may be differences between ELSPAC and ALSPAC. I think elaboration on this point is needed and clearly stating why this might impact results.

5. The authors mention that previous studies have observed some sex differences. Given the results of these previous studies, why were no interaction effects with sex tested?

6. The paragraph spanning pages 17-18 (lines 317-339) needs to be more cohesive in the message that it is trying to convey. I think it needs to remind the readers that birth weight/length can be an index of prenatal stress impacting the prenatal environment and consequently prenatal development. The authors could then discuss that this may impact brain development, due to the foetus adapting to the stressful environment, and this consequently plays a role in how the child responds and behaves in the postnatal environment. Essentially, I think the paragraph could more clearly lay out the possible pathway between prenatal stress to birthweight to child behaviour.

7. Page 19, line 363 – change “did not” to “have not”.

6. PLOS authors have the option to publish the peer review history of their article (what does this mean?). If published, this will include your full peer review and any attached files.

Reviewer #1: No

Reviewer #2: No

---

## [Author Response · Author response to Decision Letter 0]

20 May 2021

Birth weight rather than birth length is associated with childhood behavioural problems in a Czech ELSPAC cohort

Lucie Ráčková et al.

Response to reviewers’ comments

Reviewer #1: 

A first major issue that emerges from my own reading of the paper concerns the methods used. Why is a binary response constructed for each behavior problem? Why is the original variable not used? I do not understand why it is necessary to dichotomize the variable with the corresponding loss of information that it implies. I think that you can estimate an OLS model. Perhaps a fixed effects model will be better in order to control the effect of unmeasured characteristics of the family (biased estimates). I would like to see the results of a such model.

The article’s essential aim was to compare the models between the Czech ELSPAC cohort and Avon ELSPAC cohort (ALSPAC). The original ALSPAC article [48] uses dichotomized version of the response variable and using the same methodology allowed for the direct comparison of the results. Nonetheless, we also conducted the analysis using continuous models, and we provide them in supplementary data (Table S1). We believe that the conclusions drawn from continuous models do not notably differ from the dichotomous models. We refer here to another study by Kuruczová et al. [42], where the analysis was performed using the method you describe.

Moreover, the decision to weight the data is not well justified. It is said that the method was selected to account for the somewhat different proportion of socio-economic characteristics in the Included and Excluded dataset. I understand that there is some pattern that leads to non-response, such as that families with lower socio-economic status tend to respond less or, perhaps it is a problem of the sample….. It is necessary to better specify the use of weighting, since it is true that it reduces bias but weighted often increases variances of the estimates and its use must be well justified.

We understand this point as the concern about increased variance seems a very valid point. We did make the decision to use weights after careful consideration and comparison of qualities of unweighted and weighted models. In our models, the weighting did not seem to increase the variance of the estimates and overall increased the model quality. 

Four weighted logistic regression models were fitted, one for each behavior problem and separated by birth weight and birth length. I understand that Tables 4 and 5 show the OR for the relationship of birth weight and birth length respectively, but it does not show the significance of the rest of the variables included in each model or any measure of the statistical quality of the model to compare, for example the AIC, BIC, ...

Similarly to original ALSPAC article, our main aim was to assess the association between birth weight/height and the SDQ score. The other variables were only used as confounders. Implicitly, every p-value reported in the article is connected to a statistical hypothesis that was being tested. If we were to report all p-values from the models, the number of tested hypotheses would become quite large; therefore, multiple hypothesis testing correction would have to be used. This would, in turn, result in smaller statistical power of our study. However, we understand that some readers are interested in full model results, so we added full model results to the Supplementary materials. We regard the reported p-values as purely informative and do not discuss them in our article. We agree with your point about reporting model qualities, we added model quality measures to the full model information reported in Supplementary table S1. (lines 247-248 and 261-263, pages 13-14).

 In the other hand, subjects with full information were in the Included dataset, and all other subjects, who had at least one missing data point, were in the Excluded dataset. I understand that these two databases do not have subjects in common but the Included dataset have 1796 subjects and the Excluded 3834, and that differs from the 3311 total subjects (Table 1). Also, line 198 talks about mothers in both datasets, so, I need a better explanation.

The reason for the disagreement with the number of participants in the Table 1 (3311) and in the analyzed dataset (1796+3834) is the fact that they are based on different selection criteria. In the article, we discuss the number of participants which were reported in each ELSPAC cohort profile or in WHO reports. The aim of the Table 1 was to compare ELSPAC cohorts in relevant features, such as number of children in 7 years and in total. The analyzed dataset is based on inclusion criteria specified in the article (singletons with (≥37 completed weeks gestation with available data on birth weight and length) and therefore includes subjects that dropped out before the age of 7 years. Subsequently, we divide this dataset into Included (1796) and excluded (3834) based on data availability of SDQ and covariates. The entire analyzed dataset is used to calculate weights. The final model was subsequently fitted on Included dataset using these weights.

For this reason, we added information about the origin of numbers reported in Table 1 (lines 115-117, page 6).

Regarding the description of mothers, we changed the reported results on the mothers from included dataset only (currently on lines 223-226, page 11).

Regarding the Strengths and Difficulties Questionnaire, the explanation provided is not very clear. First it is said that it contains 25 items scored 0 to 2, what does each value correspond to? Them, these are separated into five subscales measured from 0 to 10. How is it related to the previous score? I understand that each subscale contains different items to value, isn’t it? It is necessary to improve.

Based on your suggestion, we incorporated explanation of the values 0 to 2 and clarified the relation of scores, subscales, and Total difficulties score (currently on lines 152-155, page 8).

Finally, last major issue concerns the contribution of your study with the extant knowledge. While I agree that the topic is interesting, I think that these findings do not provide anything new. Maybe it would be interesting to analyze differences between sex or between some socioeconomic characteristics of the family.

Our article aimed to compare the results of the relationship between birth weight, length and SDQ scores at the age of 7 years in two cohorts, the Czech ELSPAC and Avon ELSPAC (ALSPAC) cohort. The sex differences and socioeconomic differences in the SDQ score were not the primary focus of our article as they were previously analysed and published in [42]. We believe that the novelty and thought-provoking attributes of our study lies in the ability to compare methodologically concordant studies using cohorts from two different countries with very different socio-political background. In fact, these countries are socioeconomically dissimilar as the Czech ELSPAC cohort captures a unique period of socio-economic transition from communist to market economy.

Lastly, some lower priority comments:

- Explain why behaviour problems are analysed at age 7 and not at age 8 or 6 for example.

ELSPAC study protocol was set out to measure variables in the following time points of the children life: antenatal, 6 weeks, 6 months, 18 months, 3 years, 5 years, 7 years, 11 years, 15 years and 18 (19) years. The SDQ questionnaire was distributed along with other questionnaires in these time points and thus it could not have been distributed in any other time point. 

- The first time the acronym WHO is used is on line 46, but its definition is on line 72.

We corrected this omission, the definition of the WHO acronym is currently on the line 43, page 2.

- When we talk about association between variables, we can say significant or non-significant association with a certain risk. The term “weak association” is not adequate.

We apologize for this unfortunate choice of term and we removed any references to “weak associations” throughout the article.

- There is an error on line 238: 0,97 would have to be 0,79

The whole paragraph was deleted from the article, including the original line 238.

 

Reviewer #2:

 In the current study, the authors investigate whether anthropometric measures at birth are associated with later behavioural outcomes in childhood. While the study has some merit, I believe that some changes need to be made to the current manuscript before it is suitable for publication. These changes broadly include giving more context and background around the time period at which the data were collected and clearly articulating the stress that this may have caused to the pregnant mother and developing foetus.

I also have some concerns around the use of inverse-probability weights to account for missing data rather than measurement invariance and the interpretation of some negligible effects as significant. Detailed feedback is given below.

Abstract

1. Give the sample size here

The information on sample size was added into the manuscript (line 21, page 1).

2. Page 2, Line 25 should say “the Strengths and Difficulties Questionnaire” rather than “a Strengths and Difficulties Questionnaire”.

 We apologize for the mistake and we corrected it (line 21, page 2).

Introduction

1. Page 2, line 43 – what is meant by “evolutionary” forces regarding nutrition and stress? An explanation of this is needed here.

According to the cited authors, Wadhwa and Federenko, in the development of individual living organisms, the two major forces act and shape his development and fitness – the nutrition as a source of energy substrate and stress as a physical or psychological challenges or threats to homeostasis to which an individual must adapt. We thus included this specification and explanation in the sentence (line 39-40, page 2).

2. On page 3, lines 66 – 68 – the authors state that some studies focus on exploring behavioural problems in children belonging to the full birth weight spectrum, but then say that the studies differ in methodology, sample size and analytical approach. What is the extent of the differences and why might it pose a problem? It’s quite common for studies to vary in methodology, sample size and even analyses, but a robust effect should be apparent across sound methods. For example, one study may have 200 participants, and another may have 20,000. I would still consider the sample size of 200 acceptable if other methods were sound, and not necessarily a limitation. Essentially, this section is quite vague and needs more information.

Thank you very much for this comment. The major points we attempted to make was that it is very difficult to follow the effects of low and very low birth weight and to compare them with the effects in the full-term newborns across studies, due to heterogenous methodology as well as heterogenous effects of the primary as well as confounding variables. The primary strength of the proposed paper is that it compares two quite different populations but is using very similar, or almost identical, methodology of data sampling. 

3. On page 4, the authors note that the ELSPAC study centres all employ a common methodology. What exactly is meant by this – was the recruitment procedure the same? The questions asked the same?

Based on this question, we included more specific information on common methodology (lines 75-85, page 4).

4. On page 4, the authors also mention that the aim of the study was to provide data free of recall bias. How exactly was this bias reduced? For example, the SDQ still requires some level of recollection. Was the study only asking about events in the 2 weeks prior, month prior, etc.? Were women recruited during pregnancy rather than postnatally?

The recollection periods in ELSPAC cohorts were dependent on the questions asked. In case of SDQ, the mothers received the questionnaire in the year of 7th birthday of the child and the questions asked for information on behaviour of the offspring during past 6 months which is in accordance with the SDQ questionnaire methodology. In this referred section, we wanted to introduce the reader to the ELSPAC cohorts in general. Because the information of SDQ questionnaire is very specific, we included it in the Materials and methods section, subsection of Behavioural problems (line 153, page 8).

5. The authors mention that the Czech Republic, which the cohort was from, was a post-communist country undergoing a period of economic transition at the time. I think some more context around the importance of this period is needed and why it might be relevant to child development. Was it a period of stress for the mothers? This needs to be clearly articulated, and the context will be informative for non-European readers.

A dramatic change in reproductive behavior, family formation and living arrangements took place in the Czech Republic over the 1990s. After the fall and demise of the totalitarian regime in 1989, a complex transformation of the previous patterns of family life, reproduction and economic aspects of family wellbeing has been in progress in the Czech Republic. Since the mid-1990s, Czech demographers have been engaged in a lively debate about whether the second demographic transition has been taking place in the Czech Republic. This information was added to the text (lines 92-99, pages 4-5). 

6. The last sentence on page 4 (“we then compare the results with outcomes obtained using the concurrent ELSPAC cohort from United Kingdom, referred to as ALSPAC (33) as well as other studies employing Strength and Difficulties Questionnaires (SDQs) to measure behavioural problems”) sounds like data from ALSPAC and other studies are being analysed and compared to ELSPAC data within the results. I don’t think this sentence is necessary, as it is to be expected that these comparisons would be made within the discussion. Rather, I would suggest stating hypotheses instead (perhaps based on what some of these studies have found).

We rephrased the last sentence to reflect our aims better (lines 105-109, page 5).

Methods

1. The authors note at the start of page 6 that detailed information about the study is available elsewhere. While detailed information certainly isn’t expected, I still think that a brief description of the study is needed. E.g. was the cohort recruited before birth? Is it representative of other births in the Czech Republic? The authors also mention “ethical issues” – is this referring to ethics approval for the study, or were there specific ethical concerns around the study’s methodology/data collection/handling of participants?

 We apologize for the omission and we include this information in the text (lines 127-130, page 7).

2. Page 7, line 150 – give a reference for the International Standard Classification of Education.

 We included a citation in the text (line 170, page 8).

3. In the statistical analysis section, I would recommend that the authors give the Ns for the included and excludes subjects.

 We included this information in the text (lines 182-183, page 9).

4. What was the rationale for the use of inverse probability weighting (IPW) rather than multiple imputation (MI) to account for the impact of missing data? While IPW does adjust for differences between those included and those excluded in the sample, MI has the benefit of accounting for missing data while not reducing sample size. A combined approach could also be used here.

We have decided to use IPW because we find it simpler and more transparent. The SDQ scoring system allows for some amount of missing data, so randomly missing values in SDQ questionnaire were dealt with thanks to the scoring methodology. Unfortunately, ELSPAC suffers from drop-out effect and the majority of missing data was caused by it. We have considered various Multiple imputation techniques for the ELSPAC dataset in the past and from our experiences with the data, we were not able to come up with MI method that one that worked well with this part of ELSPAC data. We believe that MI is more helpful in with data randomly missing throughout the time and less helpful in situations where data is missing from one time point onwards.

Results

1. Page 11 – were the differences between boys and girls tested statistically? If so, indicate that these were statistically significant p-values. If not, then I’m not sure we can really say that males and females scored differently without statistical testing.

The differences in the SDQ score between boys and girls were not the primary focus of this article. These differences in ELSPAC cohort were analysed and discussed in another article [42]. We therefore deleted the comparison of scores between boys and girls on the page 12.

2. I would recommend indicating significant effects in the tables, particularly Tables 4 and 5. Perhaps bolding the OR and C.I. numbers for significant effects.

 Thank you for the suggestions, we modified the representation of data accordingly and bolded significant results (tables 4 and 5).

3. For the birthweight models, I don’t see the value in interpreting effects where the confidence interval contains 1, even if considered weak. My concern is that it can be misleading – indicating a more meaningful effect than what is actually present.

Thank you for this point, we understand that our original presentation is misleading. Our original motivation was to compare all results (even non-significant) to the ALSPAC article. We believed that in this specific situation, comparing the direction of association and odds ratios is still valuable. However, as you correctly point out, these results should not be presented in methods as associations.

Discussion

1. As stated with regard to the results, the effects were very weak for models with conduct problems and prosociality as the outcome, with the C.I.s containing 1. Interpreting these models as significant is a bit misleading, suggesting that there may be a more meaningful effect.

 As mentioned in the answer above, our original motivation was to compare the Central-European results with the observations from the ALSPAC article. Due to lower statistical power of our study (~1700 subjects compared to ~4800 in ALSPAC study), our analysis was likely unable to detects associations with lower effect size. Therefore, if the association was significant in the ALSPAC article, we considered beneficial to compare its point-wise estimate of odds-ratio, even when it was non-significant in our analysis. Naturally, this should have been done with more caution to avoid inappropriate or misleading conclusions.

2. Page 15, line 269 – what is meant by “genetic environment”?

 We apologize for the mistake, “genetic pool” should have been used instead of “genetic environment”. The correct term was inserted.

3. Page 15, lines 274-275 – the statement “though this was no longer apparent in our birth weight model adjusted for birth length” isn’t needed, as it is redundant given the sentence that follows.

 We deleted the redundant sentence.

4. For the first two sentences on page 16, it’s not clear how exactly this relates to why there may be differences between ELSPAC and ALSPAC. I think elaboration on this point is needed and clearly stating why this might impact results.

While the Czech ELSPAC and ALSPAC cohorts have same enrolment process and same methodology, the of course differ in the populations included – one is from the Central Europe and the other is from the United Kingdom at the island on north-west Europe. The possible differences between cohorts may be therefore both due to various ethnicity and geographical origin as well as due to different socioeconomic and political background in both investigated countries. The inter-population variability could have biased results – hypothetically, if one country would have naturally lighter children, then these might not exhibit behavioural problems which were connected to children with lower birth weight in other cohort. Some studies have found that the children growth parameters differ between European nations. However, other authors stated that the inter-population differences in birth weight and length under mothers normal condition should been minimal. Therefore, we concluded that we can rule out the inter-population influence on birth weight and length as not significant and that we should focus on other interpretations of results.

5. The authors mention that previous studies have observed some sex differences. Given the results of these previous studies, why were no interaction effects with sex tested?

Exploring interaction effects with sex is out of scope for this study. Firstly, our main aim was to explore association between SDQ and birth weight/length and compare results with the ALSPAC study [48], which did not explore these interactions. Secondly, due to sample size limitation, adding and testing interactions would decrease the statistical power of our analysis.

6. The paragraph spanning pages 17-18 (lines 317-339) needs to be more cohesive in the message that it is trying to convey. I think it needs to remind the readers that birth weight/length can be an index of prenatal stress impacting the prenatal environment and consequently prenatal development. The authors could then discuss that this may impact brain development, due to the foetus adapting to the stressful environment, and this consequently plays a role in how the child responds and behaves in the postnatal environment. Essentially, I think the paragraph could more clearly lay out the possible pathway between prenatal stress to birthweight to child behaviour.

We edited this paragraph according to the recommendations (lines 326-346, pages 17-18).

7. Page 19, line 363 – change “did not” to “have not”.

We apologize for the mistake and we corrected it. (line 370, page 19).

---

## [Decision Letter · Decision Letter 1]

9 Jun 2021

Birth weight rather than birth length is associated with childhood behavioural problems in a Czech ELSPAC cohort

PONE-D-21-02707R1

Dear Dr. Bienertová-Vašků,

We’re pleased to inform you that your manuscript has been judged scientifically suitable for publication and will be formally accepted for publication once it meets all outstanding technical requirements.

Kind regards,

Clive J Petry, PhD

Academic Editor

PLOS ONE

Additional Editor Comments (optional):

I believe that the various revisions made have improved this manuscript and that it is now acceptable for publication.

Reviewers' comments:

Reviewer's Responses to Questions

**Comments to the Author**

1. If the authors have adequately addressed your comments raised in a previous round of review and you feel that this manuscript is now acceptable for publication, you may indicate that here to bypass the “Comments to the Author” section, enter your conflict of interest statement in the “Confidential to Editor” section, and submit your "Accept" recommendation.

Reviewer #2: All comments have been addressed

2. Is the manuscript technically sound, and do the data support the conclusions?

Reviewer #2: Yes

3. Has the statistical analysis been performed appropriately and rigorously? 

Reviewer #2: Yes

4. Have the authors made all data underlying the findings in their manuscript fully available?

Reviewer #2: Yes

5. Is the manuscript presented in an intelligible fashion and written in standard English?

Reviewer #2: Yes

6. Review Comments to the Author

Reviewer #2: The authors have addressed all comments. My main recommendation would be to proofread the manuscript, as there are some grammatical errors, particularly in some of the revised content. My other suggestion would be to rephrase the term "data free of recall bias" on page 4 line 88 to "data with reduced recall bias".

7. PLOS authors have the option to publish the peer review history of their article (what does this mean?). If published, this will include your full peer review and any attached files.

Reviewer #2: No

---

## [Editor Report · Acceptance letter]

14 Jun 2021

PONE-D-21-02707R1 

Birth weight rather than birth length is associated with childhood behavioural problems in a Czech ELSPAC cohort 

Dear Dr. Bienertová-Vašků:

I'm pleased to inform you that your manuscript has been deemed suitable for publication in PLOS ONE. Congratulations! Your manuscript is now with our production department. 

Kind regards, 

on behalf of

Dr. Clive J Petry 

Academic Editor

PLOS ONE